# Digital Microinterventions in Nutrition: Virtual Culinary Medicine Programs and Their Effectiveness in Promoting Plant-Based Diets—A Narrative Review

**DOI:** 10.3390/nu17203310

**Published:** 2025-10-21

**Authors:** Virág Zábó, Andrea Lehoczki, János Tamás Varga, Ágnes Szappanos, Ágnes Lipécz, Tamás Csípő, Vince Fazekas-Pongor, Dávid Major, Mónika Fekete

**Affiliations:** 1Institute of Preventive Medicine and Public Health, Semmelweis University, 1089 Budapest, Hungary; zabo.virag@semmelweis.hu (V.Z.); ceglediandi@freemail.hu (A.L.); lipecz.agnes@semmelweis.hu (Á.L.); csipo.tamas@semmelweis.hu (T.C.); pongor.vince@semmelweis.hu (V.F.-P.); major.david@semmelweis.hu (D.M.); 2Fodor Center for Prevention and Healthy Aging, Semmelweis University, 1089 Budapest, Hungary; 3Doctoral College, Health Sciences Division, Semmelweis University, 1089 Budapest, Hungary; 4Department of Pulmonology, Semmelweis University, 1083 Budapest, Hungary; varga.janos.tamas@semmelweis.hu; 5Heart and Vascular Center, Semmelweis University, 1122 Budapest, Hungary; drszappanos@gmail.com; 6Department of Rheumatology and Clinical Immunology, Semmelweis University, 1085 Budapest, Hungary

**Keywords:** culinary medicine, virtual cooking programs, digital health, plant-based diet, microintervention, narrative review, behavior change, SMS, email, mobile application, hybrid community programs

## Abstract

Background: Plant-based diets are associated with reduced risk of chronic diseases and improved health outcomes. However, sustaining dietary changes remains challenging. Digital interventions—including virtual culinary medicine programs, web-based nutrition coaching, SMS and email reminders, mobile application–based self-management, and hybrid community programs—offer promising strategies to support behavior change, enhance cooking skills, and improve dietary adherence. These approaches are relevant for both healthy individuals and those living with chronic conditions. Methods: We conducted a narrative review of studies published between 2000 and 2025 in PubMed/MEDLINE, Scopus, and Web of Science, supplemented with manual searches. Included studies comprised randomized controlled trials, quasi-experimental designs, feasibility studies, and qualitative research. Interventions were categorized by modality (SMS, email, web platforms, mobile apps, virtual culinary programs, and hybrid formats) and population (healthy adults, patients with chronic diseases). Outcomes examined included dietary quality, self-efficacy, psychosocial well-being, and program engagement. Results: Most studies reported improvements in dietary quality, cooking skills, nutrition knowledge, and psychosocial outcomes. Virtual cooking programs enhanced dietary adherence and engagement, particularly among individuals at cardiovascular risk. Digital nutrition education supported behavior change in chronic disease populations, including patients with multiple sclerosis. SMS and email reminders improved self-monitoring and participation rates, while mobile applications facilitated real-time feedback and goal tracking. Hybrid programs combining online and in-person components increased motivation, social support, and long-term adherence. Reported barriers included limited technological access or skills, lack of personalization, and privacy concerns. Conclusions: Virtual culinary medicine programs and other digital microinterventions—including SMS, email, web, mobile, and hybrid formats—are effective tools to promote plant-based diets. Future interventions should focus on personalized, accessible, and hybrid strategies, with attention to underserved populations, to maximize engagement and sustain long-term dietary change.

## 1. Introduction

Chronic, non-communicable diseases—including cardiovascular disease, type 2 diabetes, chronic kidney disease, and certain cancers—are leading causes of mortality and morbidity worldwide [1,2,3]. According to estimates by the World Health Organization (WHO), these conditions account for approximately 74% of all deaths and impose a substantial economic and societal burden on healthcare systems and communities [4]. Dietary factors play a crucial role in the prevention and progression of these diseases [5,6,7,8,9,10]. The causes of diet-related diseases include poor nutritional choices, limited access to healthy foods, socio-economic factors, and lack of education or skills regarding meal preparation. The consequences extend beyond health outcomes, such as obesity, diabetes, and cardiovascular complications, to social and economic impacts, including increased healthcare costs and reduced quality of life.

Plant-based diets—emphasizing the consumption of whole grains, vegetables, fruits, legumes, and nuts [11,12,13,14]—have been shown to reduce the incidence and severity of chronic diseases [8,15,16,17,18,19]. Numerous prospective cohort studies and randomized clinical trials demonstrate that plant-based nutrition lowers serum LDL cholesterol and blood pressure, improves glycemic control, and positively affects body weight and inflammatory markers [20,21,22,23].

Despite growing dietary guidelines and clinical recommendations emphasizing the benefits of plant-based diets, their adoption at the population level remains limited. Barriers include inadequate cooking skills, time constraints, limited access to appropriate foods, and reduced motivation. Moreover, not all individuals can benefit from digital solutions due to limited internet connectivity, low digital literacy, or socio-economic constraints. Supporting digitally excluded populations is therefore essential and can be achieved through offline resources, community-based programs, or hybrid interventions that combine in-person and digital components.

To address these challenges, digital microinterventions—short, targeted, and easily accessible strategies—have gained increasing attention for promoting lifestyle changes and sustaining behavior modification. Virtual “culinary medicine” programs, which integrate medical knowledge with culinary skill development, enable participants to acquire the knowledge and techniques required for plant-based cooking in an interactive, online format from their own kitchens [24]. Complementary digital strategies—including nutrition coaching via mobile applications, SMS and email reminders, web-based platforms, and hybrid community programs—support behavior change by facilitating personalized goal setting, self-monitoring, and social support across both healthy individuals and populations with chronic conditions [25,26]. Understanding the reach, benefits, and limitations of these interventions is crucial for designing effective dietary programs, informing educational initiatives, and promoting social change. By addressing barriers to access, including digital exclusion, and evaluating both short- and long-term outcomes, such interventions have the potential to enhance the effectiveness of dietary recommendations, improve public health education, and foster healthier eating patterns at the community and population level.

The aim of this narrative review is to synthesize the literature published between 2000 and 2025 on how virtual culinary medicine programs and other digital microinterventions contribute to the adoption and maintenance of plant-based diets, thereby potentially reducing the burden of chronic diseases. Special attention is given to evidence of effectiveness, implementation challenges, and directions for future development.

## 2. Methods

We conducted a comprehensive literature search to identify studies published between 2000 and 2025 that evaluated the effectiveness of plant-based diets and digital nutrition microinterventions. Searches were performed in PubMed/MEDLINE, Scopus, and Web of Science databases, supplemented by manual searches to identify additional relevant articles through reference screening. Keywords included “culinary medicine,” “virtual cooking program,” “digital health intervention,” “telehealth intervention,” “plant-based diet,” “dietary behavior,” and “behavior change,” combined using Boolean operators.

Inclusion criteria were as follows: (1) studies conducted in adult populations, including randomized controlled trials (RCTs), quasi-experimental studies, feasibility studies, and qualitative research; (2) interventions aimed at promoting plant-based diets, with a particular focus on improving cooking skills, nutritional self-efficacy, and behavior change; (3) publications in English; and (4) full-text availability. Exclusion criteria included animal studies, pediatric populations, weight-loss–only programs, and studies without a digital or virtual component. The literature search yielded 1864 records (PubMed/MEDLINE: 452; Scopus: 792; Web of Science: 620). After removing duplicates and irrelevant studies, 213 records were screened in full text, of which 60 studies met the inclusion criteria.

Selected studies were analyzed using a narrative synthesis approach, highlighting intervention type, target population, and outcomes. Digital interventions encompassed a broad spectrum, including virtual cooking programs, online nutrition coaching, mobile applications, SMS and email reminders, web-based platforms, and hybrid community programs, implemented among both healthy adults and individuals with chronic conditions. Outcomes of interest included diet quality, cooking skills, nutritional self-efficacy, and psychosocial well-being.

The narrative analysis allowed for comparison across heterogeneous research domains and methodologies, enabling identification of key trends, evidence of effectiveness, and frequently reported implementation challenges. This methodology aimed to provide a comprehensive overview of the role of diverse digital microinterventions in promoting and sustaining plant-based diets, considering the variety of intervention formats and target populations.

## 3. The “Food Is Medicine” Movement: Nutrition Prescribed as Treatment

The Food Is Medicine (FIM) movement in healthcare is founded on the principle that nutrient-dense foods can be as important for disease prevention and management as pharmaceutical treatments [27]. The most commonly implemented forms include medically tailored meals, produce prescriptions, and medically tailored groceries [28].

The primary aim of FIM interventions is to reduce food insecurity and improve outcomes related to diet-associated chronic diseases (e.g., diabetes, hypertension, cardiovascular conditions) while demonstrably lowering healthcare costs [27]. Simulation models suggest that, when combined with public nutrition support programs, FIM approaches could achieve savings in the billions for populations with diabetes [28].

Comprehensive analyses indicate that FIM interventions, such as prescription-based fruit and vegetable programs, significantly improve food security, diet quality, glycemic control, blood pressure, body weight, self-management, and cost-effectiveness. However, the available evidence often derives from quasi-experimental designs, highlighting the need for larger randomized trials [29].

Within the FIM framework, microinterventions—short, targeted lifestyle interventions capable of eliciting measurable changes in dietary behaviors—are gaining increasing attention. Virtual “culinary medicine” programs, which combine cooking skills development with dietary education, have shown particular promise by enhancing participants’ cooking competence, self-efficacy, and motivation to adopt a plant-based diet in an interactive, home-accessible format [30,31] (Figure 1).

Randomized and pilot studies demonstrate that online culinary programs improve diet quality, increase fruit and vegetable intake, and reduce cardiometabolic risk factors. For example, a 9-week virtual plant-based culinary medicine intervention significantly increased participants’ diet quality scores and skin carotenoid levels, objectively confirming the dietary impact [32].

Digital health supports—such as recipe and meal-planning applications, grocery purchasing incentives, and online cooking workshops—extend microintervention accessibility to low-income and marginalized populations. These solutions operationalize the FIM philosophy, promoting plant-based dietary adoption and contributing to chronic disease risk reduction [29,33].

## 4. Digital Nutrition Interventions

### 4.1. Online Cooking Courses/Culinary Medicine

Culinary medicine programs—integrating medical knowledge with culinary skill development—represent a promising approach for promoting wider adoption of plant-based diets. With the proliferation of digital technologies, previously in-person CM education is increasingly delivered virtually, allowing broader reach and cost-effective implementation [32]. Virtual CM programs offer dual benefits: they facilitate the practical adoption of plant-based diets in daily life while positively influencing participants’ psychological well-being and quality of life. This is particularly important for the prevention and management of chronic diseases, as sustained dietary changes require both motivation and psychosocial support in addition to nutrition knowledge [32].

### 4.2. Digital Nutrition Education

Advances in digital health technologies have created significant opportunities for nutrition education in recent years. Online programs, mobile applications, web-based platforms, asynchronous learning modules, email campaigns, and SMS-based messaging enable participants to access evidence-based nutrition information and behavior change tools without geographic or temporal constraints. Digital education provides a personalized, flexible, and interactive learning environment, supporting sustainable improvements in dietary behaviors by leveraging individual motivation and skill development [34].

Numerous studies have demonstrated that digital nutrition programs positively influence diet adherence, food choices, nutrition knowledge, and chronic disease risk factors [35,36,37]. For example, online cooking courses and culinary medicine programs allow participants to acquire not only theoretical knowledge but also practical skills that support the integration of plant-based foods into everyday diets. Other interventions, such as smartphone apps, web-based self-monitoring platforms, and SMS reminders, leverage behavior change theories (e.g., self-regulation, goal setting) to facilitate self-monitoring, goal tracking, and long-term engagement [38,39].

Digital nutrition education is particularly valuable for populations with limited access to healthcare or where traditional in-person education faces logistical or cost-related barriers. Interactive, multimodal content—including videos, infographics, microlearning units, and recipe databases—enhances engagement, improves the user experience, and supports long-term sustainability. At the same time, the effectiveness of digital interventions is influenced by user engagement, technological access, and personalization, which remain key areas for future research and development [40,41].

### 4.3. Conceptual Framework of Digital Nutrition Interventions

Digital health interventions—including online cooking courses, asynchronous learning programs, mobile applications, web platforms, email-based modules, and simpler technologies such as SMS—can complement each other in promoting broader adoption of plant-based diets and preventing chronic diseases [42]. Common features of these interventions include multi-level support for behavior change:Knowledge and literacy enhancement: Interactive content, educational videos and expert materials improve nutritional literacy.Motivation reinforcement: Goal setting, feedback, and social support sustain engagement.Skill development: Culinary medicine programs provide practical cooking skills, while short digital content (e.g., brief videos, app-based tips) facilitates home adaptation.Accessibility increase: Asynchronous online platforms, mobile technologies (apps, SMS), and web-based resources lower barriers to entry, especially for individuals facing geographic or temporal constraints.

The underlying mechanism aligns with the COM-B model (Capability, Opportunity, Motivation→Behavior): digital interventions enhance participants’ capabilities (knowledge, skills), opportunities (access, support), and motivation (internal and external incentives), ultimately leading to positive, sustainable dietary changes (Figure 2). As a result, nutritional behaviors improve, adherence to plant-based diets increases, and chronic disease risk is reduced.

## 5. Summary of Previous Research and Findings

### 5.1. Promoting Plant-Based Diets in Healthy Populations

Digital interventions in healthy populations have frequently aimed to increase plant-based dietary intake, particularly fruit and vegetable consumption. Web-based programs [43,44,45,46,47,48,49] have been predominantly implemented among adults and youth, demonstrating significant short- and long-term improvements in dietary behaviors. The effectiveness of these interventions was strongly influenced by participant engagement and activity levels [49,50]. Mobile applications and digital self-monitoring tools have also shown promise, particularly among young adults and ethnic minority populations [51,52,53]. Gamification elements and social media integration were shown to enhance user participation, self-regulation, and adherence. Simpler digital strategies, such as email- and SMS-based microinterventions, were also effective in increasing fruit and vegetable intake [50,54].

Personalized online nutrition counseling provided additional benefits. In the Food4Me study [55] significant improvements were observed in adherence to the Mediterranean diet, while the DASH for Health program [56] was associated with favorable changes in blood pressure and body weight. Similarly, long-term, web-based, computer-tailored nutrition programs [45,47] improved fruit and vegetable consumption through cognitive and environmental feedback mechanisms.

In recent years, virtual cooking courses and Culinary Medicine programs have gained increasing prominence in preventive nutrition. These programs not only improved cooking competence and dietary quality but also positively influenced biomarkers, such as skin carotenoid status [30,31,32,57,58,59]. Further randomized trials demonstrated that online Culinary Medicine programs significantly increased Whole Plant Food Density (WPFD) scores, cooking skills, mindful control, and psychological well-being in older adults and individuals at cardiovascular risk [30,31,32,60].

In special populations—including older adults, postmenopausal women, and food-insecure groups—digital education and virtual cooking workshops directly contributed to improvements in dietary habits and quality of life [52,60,61,62,63,64]. Traditional, in-person programs have also shown positive effects, particularly in enhancing practical cooking skills and providing direct social support, offering a useful benchmark for evaluating digital approaches.

Overall, the evidence supports that a wide range of digital and virtual strategies can effectively promote plant-based diets across healthy adults, youth, and special populations, with engagement, self-regulation, and personalization being critical factors for maximizing effectiveness. These programs have been implemented in various formats, including 12 web-based studies, 7 traditional in-person programs, 3 mobile applications, 5 virtual education initiatives (such as Zoom or online cooking classes), and 1 email-based intervention, highlighting the diversity of approaches available to support dietary behavior change (Table 1, Table 2 and Table 3).

### 5.2. Digital Nutrition Interventions in Patients with Chronic Diseases

A total of 32 randomized controlled trials and clinical studies were analyzed, examining digital nutrition interventions in populations living with chronic diseases. The interventions can be classified into five main categories: (1) simple message-based programs (SMS or email reminders), (2) smartphone applications and integrated monitoring systems, (3) web-based platforms and e-coaching programs, (4) virtual culinary medicine and digital nutrition education, and (5) hybrid approaches that combine community-based and digital elements.

#### 5.2.1. SMS- and Email-Based Digital Interventions and Dietary Adherence

Several studies have shown that SMS- or email-based reminders are effective in improving dietary adherence among patients with chronic conditions. In the study by Akhu-Zaheya & Shiyab [68], SMS reminders significantly increased adherence to a Mediterranean diet in patients with cardiovascular diseases. Cicolini et al. [69] found that weekly email reminders combined with phone follow-ups enhanced fruit consumption, reduced salt intake, and improved adherence to a low-saturated-fat diet in hypertensive adults. Donaldson et al. [70] reported that personalized SMS messages increased fruit and vegetable intake, supported regular breakfast consumption, and helped meet daily step targets in overweight and obese adults. Vinitha et al. [71] observed that SMS interventions promoted healthier dietary patterns in newly diagnosed type 2 diabetes patients, while Kelly et al. [72] showed that dietitian-led phone coaching combined with tailored messages increased vegetable and fiber intake in patients with chronic kidney disease. Overall, these findings suggest that SMS- and email-based interventions are particularly effective in promoting plant-based, Mediterranean-style diets rich in fruits and vegetables while reducing salt and saturated fat intake. Their low cost and scalability further enhance the potential of these digital interventions to improve dietary adherence (Table 4).

#### 5.2.2. Digital Nutrition Interventions Through Mobile Applications and Smartphone-Based Systems

A growing body of evidence has demonstrated that smartphone-based nutrition interventions can effectively support dietary behavior change among patients living with chronic diseases, such as cardiovascular disease, type 2 diabetes, or hypertension [76,77,78,79,80] (Table 5). These interventions included stand-alone applications for self-monitoring, integrated platforms that combined dietary logging with real-time feedback, as well as personalized messaging systems [76,78].

Studies have reported improvements in adherence to healthy dietary patterns, particularly greater adherence to the Mediterranean diet, increased fruit and vegetable consumption, and reduced sodium intake [76,77,78,80]. For example, the use of the Noom Coach application among adults with type 2 diabetes improved self-reported dietary self-management and supported glycemic control [78], while the LowSalt4Life app contributed to reductions in sodium intake among hypertensive participants [80].

Overall, these mobile interventions have been associated with improvements in diet quality, enhanced use of self-management behaviors, and, in some cases, modest weight loss [76,77,79]. Their low cost, scalability, and ease of integration into daily life highlight the potential of smartphone-based tools to promote sustainable dietary improvements in populations living with chronic diseases [76,77,78,79,80].

#### 5.2.3. Web-Based Platforms and e-Coaching

Interactive web-based platforms and e-coaching interventions have shown significant potential in improving dietary knowledge, dietary patterns, and clinical outcomes in patients with chronic diseases. Multiple randomized controlled trials and interventional studies reported beneficial effects in individuals with atrial fibrillation, type 2 diabetes, chronic kidney disease, metabolic syndrome, obesity, cancer survivors, and multiple sclerosis [81,82,83,84] (Table 6).

For instance, a multi-component remote program in Spain significantly increased adherence to the Mediterranean diet and improved nutrient intake among patients post-catheter ablation for atrial fibrillation [85]. Similarly, automated or dietitian-led web-based coaching programs enhanced diet quality indices, reduced sugar and sodium intake, and improved fiber consumption in adults with type 2 diabetes and metabolic syndrome [81,82,83,84]. Improvements in dietary self-management were also reflected in favorable metabolic outcomes, including reductions in HbA1c, blood pressure, and lipid levels.

Other studies highlighted the role of digital coaching in special populations. Among patients with chronic kidney disease, online or telehealth interventions promoted reductions in sodium intake, increased vegetable and fiber consumption, and supported safe dietary management without adverse biochemical effects [72,86]. In obesity management, web-based coaching combined with phone calls and text messaging enhanced adherence to dietary goals and produced clinically relevant weight loss [87]. In cancer survivorship, a web-based self-management program improved fruit and vegetable intake, diet quality, physical activity, and quality of life [88]. Furthermore, among people with multiple sclerosis, an asynchronous nutrition education program increased nutrition literacy, food literacy behaviors, and was well accepted, while qualitative evaluation confirmed improvements in motivation, capability, and opportunity to change dietary behaviors [89,90].

Taken together, web-based platforms and e-coaching programs consistently demonstrate improvements in diet quality, nutrition literacy, and health-related behaviors, with additional benefits for weight management, quality of life, and cardiometabolic risk factors. Their scalability, low cost, and adaptability across diverse populations highlight their potential as sustainable dietary interventions in chronic disease management.

**Table 6 nutrients-17-03310-t006:** Summary of studies on web-based platforms and online nutrition coaching.

Author, Year, Country	Sample Demographics	Timing of Outcome Assessment	Method of Nutrition Assessment	Type of Digital Intervention	Nutritional Outcomes	Other Outcomes
Goni et al., 2020, Spain [85]	720 adults post-catheter ablation for atrial fibrillation (365 intervention, 355 control); recruited from 4 hospitals	Baseline, 12, 24 months	14-item MEDAS (phone-administered), semi-quantitative FFQ (dietitian-administered)	Multi-component remote program (website, app, printed materials, quarterly phone calls) with menus, tips, education, self-assessment tools	↑ Mediterranean diet adherence; ↑ fruits, olive oil, whole grains, legumes, nuts, fish, white meat; ↓ refined cereals, red/processed meat, sweets; ↑ fiber and omega-3, ↓ carbohydrates and saturated fats	High retention (95.6% at 12 mo, 94.4% at 24 mo); improved lifestyle, PA, QoL, biomarkers; atrial tachyarrhythmia monitored
Hansel et al., 2017, France [81]	120 adults (18–75 y) with T2DM and abdominal obesity; mean BMI 33, mean HbA1c 7.2%; 67% women	Baseline, 16 weeks	24 h dietary recall via web-based module	Fully automated web-based e-coaching program (ANODE) with diet/PA self-monitoring, nutritional assessment, menu generator, PA prescription	↑ DQI-I (+4.55 vs. −1.68; *p* < 0.001)	↓ weight (−2.3 vs. −0.4 kg), ↓ waist circumference (−3.1 vs. −0.9 cm), ↓ HbA1c (−0.4% vs. −0.1%); ↑ VO_2_ max (NS)
Abu-Saad et al., 2019, Israel [82]	50 overweight/obese Arab adults (40–62 y) with poorly controlled T2DM; 25 per arm (I-ACE vs. SLA)	Baseline, 3, 6, 12 months	FFQ, PA questionnaire, anthropometry	I-ACE software: interactive lifestyle assessment and education tool delivered during 4 dietitian-led in-person sessions	↑ DM-related dietary knowledge; ↓ added sugar intake (−2.6% TEI); ↑ fiber intake (+2.7 g/1000 kcal)	Trends toward ↑ PA; ↓ HbA1c in both groups
Green et al., 2014, USA [91]	101 adults (35–69 y); BMI> 26; elevated BP; Framingham 10-y CVD risk 10–25%	6 months	3-day food diary, self-reported fruit/vegetable intake, PA questionnaire	Dietitian-led web-based intervention: initial in-person visit, personalized plan, BP monitor, scale, pedometer, secure messaging	↑ fruit and vegetable intake; adoption of DASH diet	↓ weight (net −3.2 kg), improved BP control (NS), reduced CVD risk (trend), high satisfaction
Humalda et al., 2020, The Netherlands [86]	99 adults with CKD (stages 1–4) or kidney transplant; mean eGFR 55 ± 22; sodium intake > 130 mmol/d	3 months (end of intervention), 9 months (post-maintenance)	24 h urinary sodium excretion	Web-based self-management program (SUBLIME): interactive diary, self-monitoring, goal setting, motivational e-coaching, group sessions	↓ sodium excretion (−24.8 mmol/d vs. control; *p* = 0.049) at 3 months (effect attenuated by 9 months)	↓ SBP (140→132 mmHg at 3 months); QoL, proteinuria, costs, self-management assessed (no long-term differences)
Kelly et al., 2020, Australia [72]	80 adults with stage 3–4 CKD (mean age 62 ± 12; 64% male)	3 months, 6 months	AHEI and exploratory dietary measures	Telehealth coaching: dietitian phone calls (biweekly, 3 mo) + tailored text messages, followed by 3 mo text-only support	No change in overall AHEI; improvements in vegetable, fiber, and core food group intake	↓ weight; no effect on BP; intervention safe, no adverse events
Lewis et al., 2019, Australia [87]	61 adults with class III obesity (BMI > 40), enrolled in public obesity management service	Baseline, 4, 8 months	Self-monitored dietary goals and behavior tracking	Telephone and SMS adjunct to standard care: monthly calls (10–30 min) + 3 texts/week	Improved dietary goal adherence	↓ weight (−4.87 kg vs. +0.38 kg); ↑ self-efficacy, treatment regulation, adherence
Ramadas et al., 2018, Malaysia [84]	128 adults with T2DM (HbA1c ≥ 7.0%); literate in English/Malay; internet access	Baseline, 6 months, follow-up	DKAB and DSOC questionnaires	Web-based program (myDIDeA): 6-month intervention with 12 modules, tailored dietary advice, fortnightly web access	↑ DKAB scores (post: 11.1 vs. 6.5; follow-up: 19.8 vs. 7.6); ↑ DSOC	↓ fasting glucose (7.9 vs. 8.9 mmol/L), ↓ HbA1c (8.5% vs. 9.1%)
Jahangiry et al., 2017, Iran [83]	160 adults ≥20 y with metabolic syndrome (80 intervention, 80 control)	Baseline, 6 months	Self-reported food records, dietary questionnaires; PA (MET-min/week)	Web-based program “My Healthy Heart Profile”: tailored calorie restriction, CVD risk assessment, feedback, messaging	↓ cholesterol (−88.4 vs. −8.3 mg/day), ↓ calories (−430 vs. −393 kcal/day), ↓ sodium (1337 vs. 1342 mmol/day); ↑ PA (moderate PA +260 vs. +102 MET-min/week)	↑ HRQoL (general health, vitality); improvements in anthropometry, CVD risk factors
Lee et al., 2014, South Korea [88]	59 breast cancer patients (stage 0–III) post-curative surgery; completed primary treatment within 12 months	Baseline, 12 weeks	3-day dietary recall; DQI	Web-based self-management program (WSEDI) using TTM strategies (assessment, education, action planning, SMS feedback)	↑ fruit/vegetable intake (≥5 servings/day); ↑ diet quality (DQI)	↑ aerobic exercise (≥150 min/week); ↑ HRQoL; ↓ fatigue; ↑ self-efficacy
Russell et al., 2024, Australia [89]	Adults with MS (≥18 y); recruited via MSWA channels; English-speaking	Baseline, 9 weeks	Online surveys: DHQ, CNLT, FLBC	Online program “Eating Well with MS”: 7 asynchronous modules + mailed resources (recipes, workbook)	↑ DHQ (dietary habits), ↑ CNLT (nutrition literacy), ↑ FLBC (food literacy behaviors)	Feasibility: high recruitment (*n* = 70), 54% completion, high acceptability
Russell et al., 2024, Australia [90]	16 adults with MS (10 completed full program, 6 partial)	~1 month post-program	Qualitative analysis (interviews, COM-B framework)	“Eating Well with MS” (asynchronous modules + resources)	Reported acquisition of nutrition knowledge, improved food literacy	Identified facilitators/barriers (social support, time, motivation); COM-B mapping confirmed impact on capability, opportunity, motivation

Abbreviations: AHEI, Alternative Healthy Eating Index; ANODE, Automated Nutrition and Exercise (web-based e-coaching program); BMI, Body Mass Index; BP, Blood Pressure; CKD, Chronic Kidney Disease; CNLT, Critical Nutrition Literacy Tool; COM-B, Capability, Opportunity, Motivation—Behaviour model; CVD, Cardiovascular Disease; DHQ, Diet Habits Questionnaire; DKAB, Dietary Knowledge, Attitude, and Behaviour questionnaire; DQI, Diet Quality Index; DQI-I, Diet Quality Index-International; DSOC, Dietary Stages of Change; eGFR, estimated Glomerular Filtration Rate; FFQ, Food Frequency Questionnaire; FLBC, Food Literacy Behaviour Checklist; HbA1c, Glycated Hemoglobin A1c; HRQoL, Health-Related Quality of Life; I-ACE, Interactive Lifestyle Assessment, Counseling and Education tool; MEDAS, Mediterranean Diet Adherence Screener; MET, Metabolic Equivalent of Task; MS, Multiple Sclerosis; MSWA, Multiple Sclerosis Western Australia; NS, Not Significant; PA, Physical Activity; QoL, Quality of Life; SBP, Systolic Blood Pressure; SLA, Standard Lifestyle Advice; SUBLIME, Self-management to Reduce Salt Intake in CKD patients program; TEI, Total Energy Intake; T2DM, Type 2 Diabetes Mellitus; TTM, Transtheoretical Model; VO_2_ max, Maximal Oxygen Uptake; WSEDI, Web-based Self-Management Exercise and Diet Intervention, ↑—increase; ↓—decrease.

#### 5.2.4. Digital Education and the Benefits of Plant-Based Diets

Digital culinary medicine programs play a pivotal role in promoting plant-based dietary patterns and facilitating their practical adoption. These interventions not only enhance cooking skills but also support increased consumption of fruits, vegetables, and other plant-based foods, thereby improving overall dietary behaviors (Table 7).

The Virtual Culinary Medicine Toolkit (VCMT) developed in the United States [33] demonstrated that digital tools and educational resources effectively increased patients’ knowledge of healthy eating, particularly regarding fruit and vegetable intake and the use of whole, minimally processed foods. Similarly, a bilingual virtual culinary medicine program [92] led to significant improvements in fruit and vegetable consumption among ethnically diverse patients with type 2 diabetes, alongside enhanced cooking confidence and nutrition knowledge.

In a recent randomized crossover trial in the United States, a virtual teaching kitchen–based vegan diet intervention [93] showed that participants could maintain high adherence to plant-based dietary patterns. The program improved dietary quality while also delivering psychosocial benefits, including reduced stress and improved well-being. Evidence from Japan [94] further supports the value of dietary education and self-monitoring, where participants adopting cooking instructions achieved reduced sodium intake and increased potassium intake—primarily through plant-based food sources—resulting in favorable changes in cardiometabolic risk factors. Overall, these findings highlight that digital culinary medicine and educational interventions effectively support the adoption of plant-based diets, which are essential in the prevention and management of diabetes, hypertension, and cardiovascular diseases.

**Table 7 nutrients-17-03310-t007:** Digital Education and Virtual Culinary Medicine Programs.

Author, Year, Country	Sample Demographics	Timing of Outcome Assessment	Method of Nutrition Assessment	Type of Digital Intervention	Nutritional Outcomes	Other Outcomes
Ai et al., 2024, USA [33]	Low-income adults with type 2 diabetes (*n* not specified)	During sessions; follow-up not explicitly reported	Interactive videos, handouts; diabetes education metrics (knowledge, skills, self-efficacy)	Virtual Culinary Medicine Toolkit (VCMT): animated videos, infographics, recipes, interactive handouts; provider toolkit for standardized messaging	↑ Food literacy, cooking skills, knowledge of MyPlate, carbohydrate management, and diabetes-related behaviors	↑ Engagement, retention, self-efficacy for preparing healthy foods; ↑ perceived social support and normative beliefs
Kitaoka et al., 2013, Japan [94]	71 hypertensive men (40–75 y); 39 intervention, 32 control	Baseline and post-intervention (duration not specified)	Dietary self-monitoring; urinary sodium and potassium excretion	Self-monitoring logbook with dietary education and cooking instructions	↓ Sodium intake, ↑ potassium intake; improved sodium-to-potassium ratio	↓ DBP (93→87 mmHg, significant); ↓ SBP (149→143 mmHg, NS); no changes in control
Krenek et al., 2025, USA [93]	40 adults at risk for CVD (75% female, mean age 64.4 ± 8.6 y); ≥5% ASCVD risk; mostly college educated	Pre- and post-4-week diet interventions (crossover design)	Adherence to vegan diet; intake monitored via teaching kitchen sessions and self-report	Virtual culinary medicine teaching kitchen (8 weekly 90 min Zoom classes, group format)	Adherence to vegan diet (high vs. low EVOO); experiential cooking-based learning	↓ Perceived stress (−19%), ↓ negative affect (−13%), ↑ positive affect (+6–8%); improved energy/fatigue and HRQoL
Macias-Navarro et al., 2024, USA [92]	Adults (18–70 y), ethnically diverse, T2DM with HbA1c > 7.0; recruited from community clinics; English/Spanish speakers	Baseline and post-intervention (5 sessions)	Questionnaires (dietary intake, cooking, shopping, self-management, barriers, knowledge); EMR (HbA1c, BMI, BP)	Virtual culinary medicine program: 5 × 90 min WebEx classes; bilingual delivery; asynchronous videos, handouts, culturally adapted recipes; grocery cards provided	↑ Fruit/vegetable intake, ↑ healthy food consumption, ↑ cooking confidence, ↑ diabetes-related knowledge	Feasibility confirmed (recruitment, retention, satisfaction); trends in HbA1c, BMI, BP; ↑ diabetes self-management and self-efficacy

Abbreviations: ASCVD, Atherosclerotic Cardiovascular Disease; BMI, Body Mass Index; BP, Blood Pressure; CVD, Cardiovascular Disease; DBP, Diastolic Blood Pressure; EMR, Electronic Medical Record; EVOO, Extra Virgin Olive Oil; HbA1c, Hemoglobin A1c; HRQoL, Health-Related Quality of Life; NS, Not Significant; SBP, Systolic Blood Pressure; T2DM, Type 2 Diabetes Mellitus; VCMT, Virtual Culinary Medicine Toolkit, ↑—increase; ↓—decrease.

#### 5.2.5. Hybrid Programs Combining Community and Digital Elements

The four reviewed hybrid programs demonstrated that combining digital and community elements can improve dietary adherence and diet quality (Table 8). Alonso-Domínguez et al. [95] implemented a multifactorial intervention among adults with T2DM, including a smartphone app, a 90 min food workshop, and weekly heart-healthy walks, which significantly increased adherence to the Mediterranean diet (ΔMEDAS +2.2) and diet quality (ΔDQI +2.5), while also enhancing physical activity. Villarini et al. [96] conducted a lifestyle intervention supported by SMS reminders, resulting in a slight increase in adherence to a healthy diet, with reductions in weight, BMI, and total cholesterol, although program attendance was low. Penn et al. [97] used ongoing support via mobile text messages, emails, and newsletters to promote healthy eating and cooking skills, leading to significant reductions in weight and waist circumference, along with increased physical activity. Shahar et al. [98] implemented group-based nutrition education with visual and printed materials, which decreased waist circumference in women and maintained total cholesterol in men. Overall, hybrid programs combining community and digital support appear effective in improving dietary adherence and diet quality, while promoting participant engagement and physical activity.

## 6. Discussion

The available body of evidence clearly indicates that digital nutrition interventions can improve dietary adherence, clinical biomarkers (e.g., HbA1c, body weight, blood pressure), and patient-reported outcomes such as self-efficacy and quality of life in both healthy and clinical populations. Interventions using simple digital tools—such as SMS reminders or email notifications—effectively supported basic dietary consistency and engagement in community-based activities. More complex approaches, including smartphone applications, web-based platforms, and multicomponent digital programs, enabled intensive self-management, real-time monitoring, and personalized feedback, thereby facilitating more substantial and sustained behavior change.

Hybrid programs that combined community-based activities (e.g., cooking workshops, group exercise, face-to-face counseling) with digital components (apps, online education, SMS/email reminders) demonstrated particularly promising outcomes among both healthy adults and individuals with chronic conditions such as type 2 diabetes and metabolic syndrome. For example, interventions integrating the principles of culinary medicine with digital support improved diet quality scores, increased adherence to Mediterranean or other healthy dietary patterns, and led to clinically meaningful reductions in body weight, waist circumference, and glycemic indices.

Key findings from this review highlight that digital microinterventions—especially when combined with culinary education and behavioral support—can effectively bridge the gap between knowledge and sustained dietary behavior change. Unlike previous reviews that primarily focused on general eHealth or tele-nutrition programs, this synthesis uniquely emphasizes the intersection of plant-based dietary promotion, culinary medicine principles, and digital microintervention delivery. This innovative perspective illustrates how technology-supported, food-centered education can translate dietary recommendations into practical and sustainable everyday habits. The inclusion of diverse study designs—from feasibility studies to randomized controlled trials—enabled a comprehensive interpretation of both behavioral and clinical outcomes.

Furthermore, recent geroscience research suggests that plant-based nutrition not only plays a key role in the prevention of chronic diseases but also serves as a cornerstone of healthy aging. High-quality plant-based diets have been shown to slow biological aging by reducing oxidative stress and systemic inflammation while supporting mitochondrial function and epigenetic stability [6,7,8,9,17,19,99,100,101,102,103,104,105,106,107,108]. Consequently, plant-based dietary patterns may contribute to prolonged healthspan and delayed functional decline, which is particularly relevant for public health strategies targeting aging populations.

Despite these promising results, substantial heterogeneity exists among studies in terms of intervention duration, intensity, digital modality, outcome measures, and follow-up periods. This variability limits direct comparability and highlights the need for standardized protocols, validated dietary assessment tools, and long-term follow-up to fully establish the efficacy and sustainability of digital nutrition interventions. Although current evidence generally indicates favorable effects, most studies focus on short-term outcomes, leaving the long-term sustainability of behavior change insufficiently explored. More standardized methodologies, validated measurement tools, and long-term controlled studies are needed to accurately determine the effectiveness and clinical relevance of digital interventions.

Our findings contribute to the growing recognition that culinary-based digital programs offer a practical and scalable strategy for promoting plant-based diets, particularly in clinical and community settings where access to in-person education is limited. By identifying the components most consistently associated with positive outcomes—such as interactive content, feedback mechanisms, and hybrid delivery models—this review adds meaningful insight to guide the design of future interventions.

## 7. Conclusions

Our findings demonstrate that digital nutrition interventions—including virtual culinary medicine programs, mobile applications, web platforms, SMS/email reminders, and hybrid programs—effectively facilitate the adoption and maintenance of plant-based diets, fulfilling the study’s aim of evaluating their impact on dietary behavior and clinical outcomes. Evidence shows improvements in diet quality, cooking skills, nutrition knowledge, self-efficacy, and clinical measures such as HbA1c, body weight, and blood pressure across both healthy populations and individuals with chronic conditions. Hybrid approaches that integrate digital and community-based activities appear particularly effective in sustaining behavior change and engagement. Future interventions should prioritize personalization, accessibility, cultural adaptation, and integration into healthcare systems. Overall, interactive content, feedback mechanisms, and hybrid delivery models consistently support long-term dietary change, highlighting the potential of culinary-based digital programs as scalable strategies for chronic disease prevention and public health promotion.

The adaptability of these digital nutrition and culinary medicine programs to the Hungarian context warrants particular attention. Hungary is among the most rapidly aging countries in Europe and faces a disproportionate burden of diet-related chronic diseases and functional decline associated with unhealthy aging [109]. Integrating virtual culinary medicine, mobile health, and hybrid digital nutrition programs into national public health strategies—such as workplace health promotion and healthy aging initiatives coordinated by Semmelweis University [109,110]—could offer a cost-effective and scalable approach to improve dietary behaviors and prevent age-associated chronic diseases. Tailoring these interventions to local sociocultural factors and digital literacy levels may enhance engagement, support health equity, and contribute meaningfully to extending healthspan in the Hungarian population.

## Figures and Tables

**Figure 1 nutrients-17-03310-f001:**
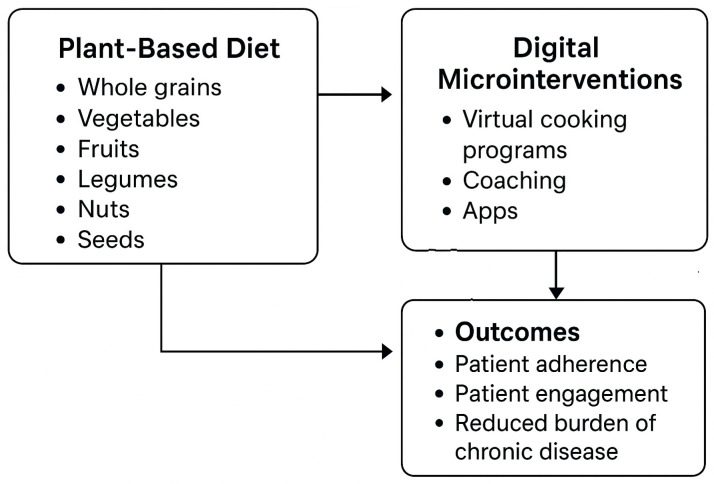
Conceptual framework linking plant-based diets, digital microinterventions, and patient outcomes. Plant-based diets (whole grains, vegetables, fruits, legumes, nuts, seeds) combined with digital microinterventions (virtual cooking programs, coaching, apps) may enhance patient adherence and engagement, thereby reducing the burden of chronic disease.

**Figure 2 nutrients-17-03310-f002:**
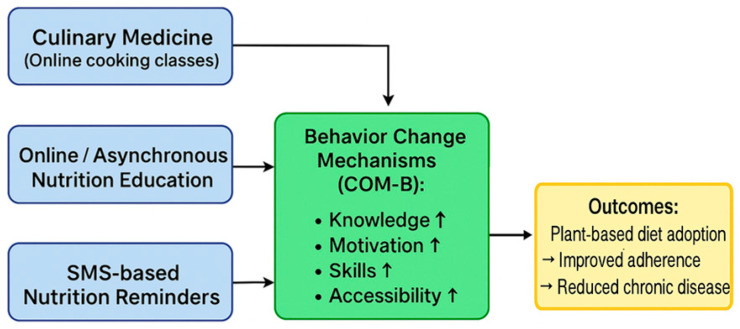
Pathways of digital nutrition interventions (culinary medicine programs, online education, and SMS-based support) in facilitating plant-based dietary adoption, conceptualized within the COM-B behavior change framework. ↑—increase/improvement.

**Table 1 nutrients-17-03310-t001:** Characteristics and outcomes of web-based digital nutrition interventions promoting plant-based diets in healthy populations.

Author, Year, Country	Sample Demographics	Timing of Outcome Assessment	Method of Nutrition Assessment	Type of Digital Intervention	Nutritional Outcomes	Other Outcomes
Alexander et al., 2010, USA [43]	*n* = 2540 adults (21–65 y), 5 health insurers, oversampled ethnic minorities	Baseline, 3, 6, 12 months	16-item NCI FFQ, 2-item short questionnaire	Web-based MENU program: (1) control, (2) tailored web, (3) tailored web + email MI	FV intake + 2 servings/day in all arms; largest increase in arm 3 (+2.8 servings, *p* = 0.05)	High satisfaction, easy scalability, good acceptability
Buller et al., 2008, USA [44]	*n* = 755 adults, 65% Hispanic, 9% Native American, 88% female, rural	Baseline, 4 months	FFQ (All-Day Screener), single-item FV question	Web-based “5 a Day, the Rio Grande Way” site (recipes, tips, community info)	FV intake ↑ (FFQ: ns; single-item: significant ↑, OR = 1.84, *p* < 0.05)	Website usage low/variable; activity associated with FV increase
Lippke et al., 2016, Germany [49]	*n* = 701 adults, mean age 38 y, 81% female, high education	Baseline (T1), 1 week (T2), 1 month (T3)	Self-report, FV servings/day, planning scales	Internet-based action and coping planning modules (vs. active and waitlist control)	FV intake ↑ (T3); planning mediated change	Engagement moderated effect (inverse U); intervention clarified by moderated mediation
Moore et al., 2008, USA [56]	*n* = 2834 adults (EMC employees + family), 26% active at 12 months	Baseline, 12 months	Self-report (weight, BP, dietary logs); DASH 24 h recall validated vs. FFQ	Web-based DASH for Health program with weekly articles, emails, self-monitoring	FV intake ↑, soda ↓, grains ↓	Overweight: −4.2 lbs; hypertensive: SBP −6.8 mmHg; dose–response with log-ins
Livingstone et al., 2016, 7 EU countries [55]	*n* = 1607 adults ≥18 y, 7 countries	Baseline, 3, 6 months	Online FFQ (157 items), MedDiet score (PREDIMED 14-point)	6-month, 4-arm RCT (general advice vs. personalized nutrition [diet/phenotype/genotype])	MedDiet score higher in PN group, highest in genotype-based PN	Baseline MedDiet score linked to BMI, physical activity; PN group showed moderate but significant improvement
Springvloet et al., 2015, The Netherlands [47]	*n* = 1349 adults (20–65 y), randomized (basic *n* = 456, plus *n* = 459, control *n* = 434)	Baseline, 9 months	Online questionnaire: FV, snacks, saturated fat, BMI, self-regulation	Web-based tailored nutrition education (basic: cognitive; plus: cognitive + environmental)	Basic: vegetable intake ↑ in low/medium educated (ES = 0.32); no effect in high-educated	Long-term effect limited; self-regulation change smaller in intervention than control
Lange et al., 2013, Germany [50]	*n* = 791; age M = 37.7 (14–79), 79% women, BMI M = 25.6, 70% college degree	Baseline & 1 week follow-up	Self-reported fruit intake (portions/day)	1 h online self-regulation intervention with volitional prompts	↑ Fruit consumption in intervention vs. control	Improved dietary planning and action control; brief intervention effective despite short duration
Tapper et al., 2014, UK [45]	*n* = 100; age M = 39, 82% female, BMI M = 27.7, 93% white	Baseline, 3, 6 months	Block Fat/Sugar/FV FFQ + BMI, WHR, HRV, IPAQ, alcohol & smoking questionnaires	Internet-based healthy eating program with 24 weekly sessions, reminders, gamified incentives	↓ Saturated fat & added sugar intake, ↑ F&V intake	Improvements in BMI, WHR, HRV; adherence monitored; incentives improved participation
Franko et al., 2008, USA [48]	*n* = 476 undergraduates, 18–24 y, 6 US universities	Baseline, post-test, 3 & 6 months	FFQ, single-item F&V servings, Nutrition Knowledge Test	MyStudentBody.com-Nutrition, interactive web program + booster session	↑ F&V intake (0.33 & 0.24 servings), ↑ nutrition knowledge	↑ Motivation, self-efficacy, social support; attitude toward exercise improved
Springvloet et al., 2015, The Netherlands [46]	*n* = 1349 adults (20–65 y), general population	Baseline, 1 month, 4 months postintervention	Self-reported FV, high-energy snacks, saturated fat	Web-based computer-tailored nutrition education (basic vs. plus with environmental feedback)	FV intake ↑ (plus version), high-energy snacks ↓ (both), saturated fat ↓ (basic)	More effective than generic info, especially in high-educated; email reminders improved engagement
Lindsay et al., 2008, UK [65]	*n* = 108 adults (50–74 y) with CHD from deprived area	Baseline, 6 months	Self-reported diet (bad foods), alcohol, exercise, smoking, mental health, social support	Password-protected health portal with weekly sessions, phone support, forums	↓ Frequency of “bad foods”	↑ Health visits; slight improvements in diet, alcohol, smoke exposure; peer support engagement
Ghammachi et al., 2022, Australia [60]	*n* = 17 young adults (18–25 y)	Pre- and post-program (4 weeks)	Online surveys: knowledge, attitudes, practices, FV intake; Facebook engagement	4-week web-based experiential program via private Facebook group (quizzes, videos, challenges)	Improved knowledge, attitudes, motivation; FV intake improved (pilot, no sig. testing)	Engagement data via Facebook; certificate of completion; prize draws

Abbreviations: BMI, body mass index; BP, blood pressure; CHD, coronary heart disease; DASH, Dietary Approaches to Stop Hypertension; EMC, EMC Corporation; ES, effect size; EU, European Union; FFQ, food frequency questionnaire; FV, fruit and vegetable(s); HRV, heart rate variability; IPAQ, International Physical Activity Questionnaire; MedDiet, Mediterranean Diet; MI, motivational interviewing; M, mean; NCI, National Cancer Institute; ns, non-significant; OR, odds ratio; PN, personalized nutrition; PREDIMED, Prevención con Dieta Mediterránea; RCT, randomized controlled trial; SBP, systolic blood pressure; WHR, waist-to-hip ratio; y, years, ↑—increase; ↓—decrease.

**Table 2 nutrients-17-03310-t002:** Summary of digital nutrition interventions using mobile applications, virtual cooking classes (Zoom teaching kitchens), and email programs to promote plant-based dietary behaviors.

Author, Year, Country	Sample Demographics	Timing of Outcome Assessment	Method of Nutrition Assessment	Type of Digital Intervention	Nutritional Outcomes	Other Outcomes
Rodgers et al., 2016, USA [51]	*n* = 43 minority female university students, BMI ≥ 25 and <21 subgroups	Baseline, 3 weeks, 10-week follow-up	Self-report (FV, SSB intake), baseline BMI	Mobile food photography + 3 daily motivational SMS	BMI ≥ 25: FV ↑; BMI < 21: fruit ↓, vegetables ↔; SSB ↔	Short-term support for healthy eating; effect differed by BMI
Nour et al., 2019, Australia [52]	*n* = 97 young adults, mean 24.8 y, 49% adhered for 4 weeks	Baseline, 4 weeks	App logging (vegetable intake) + engagement data	Smartphone self-monitoring and goal-setting app ± gamification and/or Facebook support	App usage duration associated with ↑ vegetable intake (*p* < 0.001)	Gamification/social support had no direct effect; engagement higher in persistent users
Staffileno et al., 2018, USA [53]	*n* = 26 young African American women, 18–45 y, prehypertension	Baseline, 12 weeks	6-item DASH screener, pedometer, BP, BMI	Web-based eHealth platform (12 modules, DASH vs. PA arm), mobile access + coaching	DASH group: DASH score ↑ (*p* = 0.001); large effect on FV and dairy intake	PA group: +39% steps, weight loss; engagement differed (71% vs. 48%)
Sommer et al., 2023, USA [30]	*n* = 609 peri- & postmenopausal women, mean age 58.8, 88% White	Pre- & post-intervention surveys	Self-reported weight, BMI, FFQ (FV, fish, beans, red meat, sugary drinks, grains), cooking confidence survey	NuCook virtual teaching kitchen, synchronous online classes with live cooking & nutrition discussion	↑ FV, fish, beans; ↓ red meat, sugary drinks, white grains; small weight loss	↑ Cooking self-efficacy & confidence; ↓ BMI in obese subgroup
Glickman et al., 2024, USA [57]	*n* = 360 osteopathic medical students (249 in-person, 111 virtual)	Post-course survey after 4-module culinary medicine course	Self-generated survey: knowledge (5 items), enjoyment (2 items), Likert scale	Virtual (Blackboard Collaborate) or in-person culinary medicine course	Knowledge ↑ in in-person group	Enjoyment ↑ in in-person group (Cohen’s d = 0.807); high satisfaction; reliability acceptable (Cronbach’s α: knowledge = 0.74, enjoyment = 0.77)
Charles et al., 2023, USA [31]	*n* = 80 physician assistant students, medical trainees	Pre-intervention, immediately post-intervention, 4 weeks post-intervention	Self-reported knowledge, attitudes, confidence, personal dietary behaviors	Interactive, single-session virtual curriculum (didactic + assessment/counseling + virtual culinary medicine via Zoom)	Knowledge ↑ (48.9%→78.9%, retained 75.8% at 4 weeks); FV counseling confidence ↑	Attitudes improved on diet–disease reversal; scalable teaching kitchen; engagement via Zoom breakout rooms
Krenek et al., 2025, USA [32]	*n* = 40 adults, 75% female, age 64 ± 9 y, BMI 32 ± 7, ≥5% ASCVD risk	Baseline and post each 4-week diet period	ASA-24 Automated 24 h Dietary Recall, VeggieMeter^®^ skin carotenoids	Weekly virtual vegan culinary medicine sessions via Zoom	↑ Whole Plant Food Density, diet quality, vegan adherence	↑ Cooking confidence, diet knowledge, perceived heart health control, CAFPAS; high satisfaction
Razavi et al., 2023, USA [58]	*n* = 1433 medical trainees (519 virtual culinary medicine, 914 standard nutrition), mean age 27, >50% women	Cross-sectional survey post-course	CHOP-MT survey: diet, MedDiet adherence, nutritional attitudes, competencies	Virtual culinary medicine (Zoom/WebEx), team-based cooking & discussions	MedDiet adherence ↑ (fruit intake OR 1.37)	↑ Lifestyle medicine competencies (fiber OR 4.03, T2DM prevention OR 4.69, omega fatty acids OR 5.21, MedDiet counseling OR 5.73)
Kothe & Mullan, 2014, Australia [54]	*n* = 275 university students (≥18 y)	Baseline, 30 days	Self-reported FV intake (servings/day)	30-day email intervention (daily vs. every 3-day messages)	FV intake ↑ in both groups	High-frequency messages perceived as excessive; acceptability moderated effect

Abbreviations: ASCVD, atherosclerotic cardiovascular disease; ASA-24, Automated Self-Administered 24 h Dietary Recall; BMI, body mass index; BP, blood pressure; CAFPAS, Cooking and Food Provisioning Action Scale; CHOP-MT, Children’s Hospital of Philadelphia Medical Trainee survey; DASH, Dietary Approaches to Stop Hypertension; FFQ, food frequency questionnaire; FV, fruit and vegetable(s); OR, odds ratio; PA, physical activity; SSB, sugar-sweetened beverage; T2DM, type 2 diabetes mellitus; ↑—increase; ↓—decrease; ↔—no significant change.

**Table 3 nutrients-17-03310-t003:** Summary of traditional, group-based, and hybrid interventions (cooking classes, community programs, and Zoom-delivered nutrition education) promoting plant-based dietary behaviors.

Author, Year, Country	Population & Setting	Timeline	Measures	Intervention	Main Outcomes	Additional Findings
Diallo et al., 2020, USA [63]	566 older adults (60% female, 81% African American, age 45–95, low-income)	Ongoing during program	USDA Six-Item Food Security, Lubben Social Network Scale	Healthy Meal Program: weekly congregate meals, home delivery, mobile market, 8-week Kitchen Clinic	↑ Access to vegetables, ↑ fresh produce intake	↓ Food insecurity, ↓ social isolation; strong engagement in education
Delichatsios et al., 2015, USA [66]	70 adults with ≥1 cardiovascular risk factor, primary care	17 SMA sessions over 4 years	Patient surveys (knowledge & satisfaction)	Shared Medical Appointments with cooking demos + nutrition education	↑ Nutrition knowledge, ↑ cooking skills, improved dietary strategies	High satisfaction; cost-effective; labs and meds adjusted during sessions
Kwon et al., 2015, Japan [64]	89 prefrail women ≥70 y	Baseline, 12 weeks, 6 months	Dietary variety score, cooking classes, HRQOL	Group-based exercise + nutrition program	↑ Dietary variety, ↑ HRQOL domains	↑ Handgrip, balance, walking speed
Peters et al., 2014, USA [61]	Healthy postmenopausal women, 50–72 y, BMI 18–30	Baseline, 6 m, 12 m	Monthly 24 h food recalls, adherence score	Hands-on cooking + behavioral (social cognitive theory)	Significant adoption & maintenance of diet patterns	↓ Non-adherence; psychosocial predictors important
Moreau et al., 2015, Canada [62]	144 community-dwelling adults ≥50 y	Pre- & post-8 workshops	Elderly Nutrition Screening Q., session surveys	Cooking + nutrition workshops	↑ Knowledge, confidence, ↑ intake of whole grains, F&V, milk alternatives	Confidence linked to healthier diet; autonomy unchanged
Shavit et al., 2024, Israel [59]	211 adults, plant-based MedDiet (Fixed *n* = 95, Changing *n* = 116)	Baseline, 6 w, 3 m FU, 6 m FU	Food diversity, I-MEDAS, % plant foods	Weekly Zoom sessions + digital menus	Fixed menu: sustained MedDiet adherence, ↑ plant foods	High completion (97%); variety explored; taste ratings collected
Janko et al., 2025, UK [67]	37 Seventh-day Adventists (vegan/veg/pesc.)	Pre, post, 4 w FU	25-item nutrition knowledge test, follow-up survey	Single 30 min Zoom lecture by nutrition expert	↑ Knowledge (8.5→20.0/25), ↑ supplement use	Behavior changes at 4 w; framed by Health Belief Model

Abbreviations: SMA, Shared Medical Appointment; HRQOL, Health-Related Quality of Life; MedDiet, Mediterranean Diet; I-MEDAS, Israeli Mediterranean Diet Adherence Screener; F&V, Fruits and Vegetables; FU, Follow-up, ↑—increase; ↓—decrease.

**Table 4 nutrients-17-03310-t004:** SMS- and email-based interventions to promote dietary change in patient populations.

Author, Year, Country	Sample Demographics	Timing of Outcome Assessment	Method of Nutrition Assessment	Type of Digital Intervention	Nutritional Outcomes	Other Outcomes
Akhu-Zaheya & Shiyab, 2016, Jordan [68]	160 adult outpatients with CVDs (≥18 y; excluded DM, renal, neurological disease)	Baseline, 3 months	MEDAS	SMS reminders (diet-focused) vs. placebo SMS vs. control	↑ Mediterranean diet adherence (*p* < 0.001)	↑ Medication adherence (*p* = 0.001); no effect on smoking
Dawson, 2021, Australia [73]	130 adults on hemodialysis; ≥18 y; English-speaking	Baseline, 3, 6 months	24 h recall, portion models, Foodworks	SMS (3/week; personalized 0–3 m, general 4–6 m)	No significant adherence to protein, K, P, Na guidelines; exploratory ↓ intake of these nutrients	Recruitment 48%, retention 88%; themes of acceptability; IDWG, electrolytes, binder use, QoL, healthcare use
Vinitha et al., 2019, India [71]	248 adults with newly diagnosed T2DM, mean age 43.3 ± 8.7 y	Baseline, 3, 6, 12, 18, 24 months	24 h recall, nutrient intake per NIN guidelines	2–3 SMS/week on lifestyle + medication adherence	Promoted healthier dietary patterns (details not specified)	↓ HbA1c, ↓ LDL, ↓ weight, ↓ waist, ↓ BP, ↑ QoL, ↑ PA; high acceptability
Yasmin et al., 2020, Bangladesh [74]	320 adults with T2DM (160 I, 160 C; 273 completed)	Baseline, 12–16 months	Structured interviews; anthropometry & labs	Mobile phone interactive voice calls every 10 d + 24/7 call center	↑ Adherence to dietary advice	↑ Medication & exercise adherence, ↓ tobacco use, ↑ glycemic control
Cicolini et al., 2014, Italy [69]	198 hypertensive adults, mean age 59 ± 14.5 y	Baseline, 1, 3, 6 months	Validated diet questionnaires, daily self-assessment, food group tables	Nurse-led weekly emails + phone follow-up	↑ Fruit intake, ↓ obesity prevalence, ↑ adherence to low-salt, low-fat diet	↓ BP, ↓ LDL, ↓ total cholesterol, ↓ TG, ↓ glucose; ↑ PA, ↑ med adherence, ↑ smoking cessation
Donaldson et al., 2014, UK [70]	34 obese/overweight adults (BMI ≥ 30 or ≥28 + comorbidity); post-LEAP program	Pre- and post-12 weeks	Self-reported F&V & breakfast intake; step count; QoL questionnaires	SMS twice weekly; feedback loop with practitioner	↑ F&V, ↑ breakfast intake, ↑ adherence to step goals	↓ Weight (−1.6 kg), ↓ BMI (−0.6), ↓ waist (−2.2 cm), ↑ QoL, better follow-up
Islam et al., 2021, Bangladesh [75]	236 adults with T2DM, ≤5 years since diagnosis, on oral meds	Baseline, 6 months	WHO STEPS survey + FFQ; weekly servings	Daily SMS × 6 months (diet, PA, meds, diabetes education)	No significant change in fruit/vegetable intake; ↓ sugared beverage intake (NS); ↑ sugar in tea (+0.94 tsp/w, *p* < 0.05)	HbA1c, BP, anthropometry measured; no major clinical effect; feasible
Kelly et al., 2020, Australia [72]	80 adults with stage 3–4 CKD, mean age 62 ± 12 y	3 months (end Phase 1), 6 months (end Phase 2)	AHEI; exploratory diet measures (veg, fiber, food groups)	Telehealth coaching: 3 m dietitian calls + SMS, then 3 m SMS only	No significant effect on AHEI; improved veg intake, fiber, core food groups	↓ Weight; no BP change; safe, no adverse events

Abbreviations: CVD, Cardiovascular Disease; DM, Diabetes Mellitus; MEDAS, Mediterranean Diet Adherence Screener; SMS, Short Message Service; T2DM, Type 2 Diabetes Mellitus; NIN, National Institute of Nutrition; I, Intervention; C, Control; BP, Blood Pressure; LDL, Low-Density Lipoprotein; TG, Triglycerides; PA, Physical Activity; F&V, Fruits and Vegetables; FFQ, Food Frequency Questionnaire; WHO STEPS, World Health Organization STEPwise survey; AHEI, Alternative Healthy Eating Index; QoL, Quality of Life; IDWG, Interdialytic Weight Gain; NS, Not Significant, ↑—increase; ↓—decrease.

**Table 5 nutrients-17-03310-t005:** Mobile Application and Smartphone-Based Nutrition Interventions.

Author, Year, Country	Sample Demographics	Timing of Outcome Assessment	Method of Nutrition Assessment	Type of Digital Intervention	Nutritional Outcomes	Other Outcomes
Choi, B.G. et al., 2019, USA [76]	100 cardiology patients (mean age ~57, 60% male, 20–35% CAD)	Baseline, 1, 3, 6 months	Mediterranean Diet Score (MDS)	Smartphone app with asynchronous dietary counseling (custom app by Vibrent Health), 60 min RD interaction; SOC: 2 extra face-to-face sessions at 1 and 3 months	↑ Adherence to Mediterranean diet over time in both EXP and SOC groups; no significant difference between groups	↑ Weight loss (EXP 3.3 lbs vs. SOC 3.1 lbs, *p* = 0.04); ↑ diet satisfaction over time; BP, lipids, HbA1c, CRP showed no significant differences
Allen, J.K. et al., 2013, USA [77]	*n* = 68; 78% female; 49% African American; mean age 45 ± 11 years; BMI 34.3 ± 3.9 kg/m^2^	Baseline and 6 months	3-day food records analyzed via NDSR	Smartphone app (“Lose It!”) for self-monitoring diet, exercise, weight; delivered alone or with intensive/less intensive behavioral counseling	Trends toward improved dietary intake in counseling + smartphone groups (specifics not detailed)	BMI, waist circumference, physical activity, feasibility, acceptability, adherence to intervention
Ku, E.J. et al., 2020, South Korea [78]	40 adults with T2DM (20 SC, 20 CC), aged 20–80; exclusion: cognitive impairment, inability to use smartphone, medications affecting glucose control	Baseline and 12 weeks	Dietary logging via Noom Coach app (SC group), SDSCA questionnaire (all participants)	Smartphone-based integrated online real-time diabetes care system: Noom Coach app for dietary logging, CareSens N NFC glucose meter, individualized text feedback, social network support (SC group)	↑ Self-reported dietary management (general diet, specific diet) in SC vs. CC; SDSCA scores increased in both groups	Glycaemic control: higher proportion achieving A1C < 6.5% in SC vs. CC (47.1% vs. 11.1%, *p* = 0.019); improvements in blood glucose testing and foot care; no major adverse events reported
Boels, A.M. et al., 2019, The Netherlands [79]	228 adults with T2DM, aged 40–70, on insulin ≥3 months, HbA1c > 7%; intervention *n* = 114, control *n* = 114	Baseline, 6 months follow-up, additional 3-month sustainability follow-up (intervention group)	FFQ	Smartphone app delivering unidirectional evidence-based messages on diet, physical activity, hypoglycaemia prevention, and glucose variability; frequency, topics, and duration tailored by patient (6–9 months)	Not explicitly reported yet; app targeted dietary habits and self-management behaviors	Primary: HbA1c and % achieving HbA1c ≤ 7% without hypoglycaemia; Secondary: BMI, waist circumference, insulin dose, lipid profile, BP, hypoglycaemic events, glycaemic variability, self-management (SDSCA), physical activity, health status, diabetes-dependent QoL, treatment satisfaction, cost-effectiveness, sustainability
Dorsch, M.P. et al., 2020, USA [80]	50 adults ≥ 18 y, hypertensive, iPhone users; excluded CKD, heart failure, severe HTN, insulin-treated diabetes, loop diuretics, corticosteroids, NSAIDs	Baseline and week 8	24 h urinary sodium (spot & collection), FFQ, ASA24 24-h dietary recall, sodium screener	Just-in-time adaptive mobile app (LowSalt4Life) with push notifications, geolocation-based suggestions, low-sodium food alternatives, restaurant/grocery search	↓ Sodium intake (spot urine: App −462 mg vs. No App +381 mg, *p* = 0.03; FFQ: App −1553 mg vs. No App −515 mg, *p* = 0.01)	BP: App −7.5 mmHg vs. No App −0.7 mmHg (*p* = 0.12); Self-confidence in following low-sodium diet: no significant difference

Abbreviations: CAD, Coronary artery disease; SOC, Standard of care; EXP, Experimental group; BMI, Body mass index; MDS, Mediterranean Diet Score; RD, Registered dietitian; T2DM, Type 2 diabetes mellitus; SC, Smartphone care group; CC, Conventional care group; SDSCA, Summary of Diabetes Self-Care Activities; FFQ, Food Frequency Questionnaire; ASA24, Automated Self-Administered 24 h Dietary Recall; BP, Blood pressure; HbA1c, Glycated hemoglobin; CRP, C-reactive protein; QoL, Quality of life, ↑—increase; ↓—decrease.

**Table 8 nutrients-17-03310-t008:** Hybrid Programs Combining Community and Digital Elements.

Author, Year, Country	Sample Demographics	Timing of Outcome Assessment	Method of Nutrition Assessment	Type of Digital Intervention	Nutritional Outcomes	Other Outcomes
Alonso-Domínguez et al., 2019, Spain [95]	204 adults with T2DM (25–70 yrs, mean 60.6; excluded CVD, musculoskeletal, neuropsychological disease)	Baseline, 3 months, 12 months	MEDAS questionnaire, Diet Quality Index (DQI)	Multifactorial: smartphone app (EVIDENT II) for 3 months; 90 min food workshop; weekly heart-healthy walks (5 weeks)	↑ Adherence to Mediterranean diet (ΔMEDAS +2.2 at 3 months, sustained at 12 months); ↑ Diet quality (ΔDQI +2.5 at 3 months)	↑ Physical activity (weekly walks attendance 80–90%); ↑ app engagement (days of use recorded); biomedical parameters monitored (glucose, HbA1c)
Villarini et al., 2015, Italy [96]	186 adults aged ≥45, community pharmacy volunteers	Baseline, 6 months	Self-reported adherence to healthy diet; measured anthropometrics and clinical parameters in pharmacies	Lifestyle intervention with digital support: SMS reminders for conferences, cooking classes, physical activity sessions	Slight increase in adherence to healthy diet in males; no significant changes in diet-related biomarkers	Significant reductions in weight, BMI, total cholesterol; no significant change in waist circumference, BP, fasting glucose, triglycerides; metabolic syndrome prevalence decreased non-significantly in women; session attendance low
Penn et al., 2013, UK [97]	218 adults aged 45–65 at high risk of T2D (FINDRISC ≥ 11), socioeconomically deprived area; 134 completed follow-up (61%)	Baseline, 6 months, 12 months	Self-report dietary questionnaire on fruit, vegetable, bread, milk, fat consumption; aligned with dietary advice	Ongoing support via mobile text messages, emails; newsletters with information and recipes	↑ Adherence to healthy eating; use of healthy cooking demonstrated in sessions	Weight decreased by 5.7 kg (men) and 2.8 kg (women); waist circumference decreased 7.2 cm (men) and 6.0 cm (women); PA level increased 7.9 MET h/day (men) and 6.7 MET h/day (women); high intervention acceptability and retention
Shahar et al., 2013, Malaysia [98]	47 older Malays with metabolic syndrome (60–75 yrs; 24 intervention, 23 control; 50.6% men, 49.4% women)	Baseline, 6 months	Anthropometric measurements (weight, waist); dietary counselling using culturally tailored materials	Group-based nutrition education sessions with flipcharts, booklets, placemats, cooking & exercise demonstrations (interactive visual aids, print materials)	Women: ↓ waist circumference; Men: maintained total cholesterol	Fasting blood glucose, lipid profile, BP, CRP; adherence/compliance assessed weekly then monthly

Abbreviations: T2DM, Type 2 Diabetes Mellitus, CVD, Cardiovascular Disease, MEDAS, Mediterranean Diet Adherence Screener, DQI, Diet Quality Index, SMS, Short Message Service, BP, Blood Pressure, PA, Physical Activity, CRP, C-reactive protein, ↑—increase; ↓—decrease.

## Data Availability

No new data were created or analyzed in this study.

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
