# Peer review of "Digital Microinterventions in Nutrition: Virtual Culinary Medicine Programs and Their Effectiveness in Promoting Plant-Based Diets—A Narrative Review"

_nutrients, 2025, doi:10.3390/nu17203310_

Round 1

Reviewer 1 Report

Comments and Suggestions for Authors

Dear Authors,

I carrefully read the manuscript „Digital Microinterventions in Nutrition: Virtual Culinary Med-icine Programs and Their Effectiveness in Promoting Plant-Based Diets – A Narrative Review”. The aim of this review was to synthesize the literature published between 2000 and 2025 on how virtual culinary medicine programs and other digital microinterventions contribute to the adoption and maintenance of plant-based diets, thereby potentially reducing the burden of chronic diseases. The manuscript is well structured and the findings are relevant both for academic community and general population.

However, to enhance the clarity of presented data you should pay attention to these aspects:

Methods: Please clarify the methodology by adding the number of studies you retrieved from PubMed/MEDLINE, Scopus, and Web of Science databases using the indicated keywords. How many studies you selected for further analysis?

Summary: I suggest to renamed this section. A Discussion section is more appropiate in order to provide additional information by summarizing the main results of your study. Emphasize the main findings of your research and what does it add to the subject area compared with other published articles.

I appreciated the description of the future directions and limitations of the study. You can include this information within the Discussion section.

The conclusion section should be consistent with the aim of the study, the evidence and arguments presented in the manuscript. Please structure the conclusions according to this suggestion.

Author Response

Thank you very much for your detailed and constructive feedback on our manuscript. We greatly appreciate your suggestions and are pleased to inform you that we have revised the manuscript according to your points:
We have updated the Methods section to specify the number of studies identified in PubMed/MEDLINE, Scopus, and Web of Science based on the selected keywords, as well as the number of studies included for further analysis.
The “Summary” section has been renamed “Discussion.” We have elaborated on the main findings, highlighting the key outcomes of our study and their contribution to the existing literature. We also incorporated the discussion of future research directions and study limitations into this section.The “Conclusions” section has been revised to fully align with the study objectives and the presented evidence. The conclusions are now structured logically, emphasizing the potential role of digital microinterventions and culinary programs in promoting plant-based diets and preventing chronic diseases.
Once again, we sincerely thank you for your careful review and valuable suggestions.

Reviewer 2 Report

Comments and Suggestions for Authors

It is a well-designed and written narrative review.

Author Response

Thank you very much for your kind feedback! We truly appreciate your recognition of the design and writing of our manuscript.

Reviewer 3 Report

Comments and Suggestions for Authors

Recommendations:
Please submit your manuscript using the format recommended by the publisher.
Is a 5-year review of scientific reports sufficient? Why was this review period established?
The introduction should be expanded, as the causes and effects of diet-related diseases, apps, and other digital methods supporting diets need to be further explored.
The drawbacks and limitations of such modern solutions should be analyzed in more detail. What about digitally excluded individuals? How can they be supported?
How will your observations and conclusions impact the effectiveness of diets, educational programs, and social change?

Author Response

Thank you very much for your valuable and detailed review. We would like to clarify that our manuscript does not focus only on the past 5 years, but rather provides a comprehensive overview of the literature published between 2000 and 2025, offering a broader perspective on the effectiveness of digital microinterventions and virtual culinary medicine programs.

Following your suggestions, we have expanded the Introduction to more thoroughly present the causes and consequences of diet-related diseases, the digital tools supporting dietary change, their advantages and limitations, and strategies for supporting digitally excluded individuals. We have also emphasized how our observations and conclusions may contribute to the effectiveness of diets, educational programs, and social change. Thank you for your constructive feedback.

Round 2

Reviewer 1 Report

Comments and Suggestions for Authors

Dear Authors,

Thank you for your concise and clear responses. You've made the changes according to my suggestions, so I consider that your paper is appropiate for publication in this journal.